# Dust deposition and ambient PM$_{10}$ concentration in northwest China: Spatial and temporal variability

Xiao-Xiao Zhang[1,2], Brenton Sharratt[3], Xi Chen[1], Zi-Fa Wang[2], Lian-You Liu[4], Yu-Hong Guo[2], Jie Li[2], Huan-Sheng Chen[2], and Wen-Yi Yang[2]

[1]State Key Laboratory of Desert and Oasis Ecology, Xinjiang Institute of Ecology and Geography, Chinese Academy of Sciences, Urumqi, 830011, China

[2]State Key Laboratory of Atmospheric Boundary Layer and Atmospheric Chemistry, Institute of Atmospheric Physics, Chinese Academy of Sciences, Beijing, 100029, China

[3]USDA-ARS, 215 Johnson Hall, Washington State University, Pullman, WA 99164, USA

[4]Key Laboratory of Environmental Change and Natural Disaster, Ministry of Education, Beijing Normal University, Beijing, 100875, China

*Correspondence to*: X. X. Zhang (zhangxx@ms.xjb.ac.cn)

**Abstract.** Aeolian dust transport and deposition are important geophysical processes which influence global bio-geochemical cycles. Currently, reliable deposition data are scarce in central and east Asia. Located at the boundary of central and east Asia, Xinjiang Province of northwestern China has long played a strategic role in cultural and economic trade between Asia and Europe. In this paper, we investigated the spatial distribution and temporal variation in dust deposition and ambient PM$_{10}$ (particulate matter in aerodynamic diameter ≤10 μm) concentration from 2000 to 2013 in Xinjiang Province. This variation was assessed using environmental monitoring records from 14 stations in the province. Over the 14 years, annual average dust deposition across stations in the province ranged from 255.7 to 421.4 t km$^{-2}$. Annual dust deposition was greater in southern Xinjiang (663.6 t km$^{-2}$) than northern (147.8 t km$^{-2}$) and eastern Xinjiang (194.9 t km$^{-2}$). Annual average PM$_{10}$ concentration across stations in the province varied from 100 to 196 μg m$^{-3}$ and was 70, 115 and 239 μg m$^{-3}$ in northern, eastern and southern Xinjiang, respectively. The highest annual dust deposition (1394.1 t km$^{-2}$) and ambient PM$_{10}$ concentration (352 μg m$^{-3}$) were observed in Hotan which is located in southern Xinjiang and at the southern boundary of the Taklamakan Desert. Dust deposition was more intense during the spring and summer than other seasons. PM$_{10}$ was the main air pollutant that significantly influenced regional air quality. Annual average dust deposition increased logarithmically with ambient PM$_{10}$ concentration (R$^2$≥0.81). While the annual average dust storm frequency remained unchanged from 2000 to 2013, there was a positive relationship between dust storm days and dust deposition and PM$_{10}$ concentration across stations. This study suggests that sand storm is a major factor affecting the temporal variability and spatial distribution of dust deposition in northwest China.

**Key words:** Dust deposition; PM$_{10}$; Desert; Atmospheric environment; East Asia

## 1 Introduction

Airborne dust generated by aeolian activity is an environmental concern in central and east Asia (Huang et al., 2011; Chen et al., 2014). Historically, aeolian activity and airborne dust influenced civilization along the ancient Silk Road which connected Asia and Europe (Zhang, 1984; Dong et al., 2012; Groll et al., 2013). Today, airborne dust is recognized as a factor affecting global radiation and warming (Stanhill, 2005; Carslaw et al., 2013; IPCC, 2013; Huang et al., 2009; Chen et al., 2013; Huang, et al., 2014) and air quality in distant lands (Tsoar and Pye, 1987; Xu et al., 2007; Uno et al., 2009; Li et al., 2012). Deposition of airborne dust also plays a significant role in soil formation and biological diversity in arid and semi-arid regions (Simonson, 1995; Lin and Feng, 2015; Varga et al., 2016). An understanding of atmospheric dust sources, emissions, and deposition is therefore essential to improve our knowledge of dust impact on regional air quality.

Dust in the atmosphere and its subsequent deposition are vital indicators of aeolian activity and environmental quality. Deposition has been measured directly at only a few sites, therefore, reliable dust deposition data are lacking around the world (Pye, 1987; Mahowald, et al., 1999; Prospero, 1999; Mahowald, et al., 2009; Zhang et al., 2010; Huneeus et al., 2011; Shao et al., 2011). Annual dust deposition ranges from 10 to 200 t km$^{-2}$ on continents and one to two orders of magnitude lower over oceans (Pye, 1987; Duce et al., 1991; Ginoux, et al., 2001; Ginoux, et al., 2012). It's estimated that annual average dust deposition rate upon desert areas ranged between 14-2100 t km$^{-2}$ (Zhang, et al., 1997). Observations of dust deposition have been made over deserts with an enhanced awareness of its significance. Table 1 lists observations of dust deposition in desert regions and other regions of the world prone to aeolian activity. According to these observations (Table 1), dust deposition is high in Asia with an annual deposition of 8365 t km$^{-2}$ in the Aral Sea Basin (Wake and Mayewski, 1994; Groll et al., 2013). Recent investigations suggest that the intensity of dust deposition is closely related to weather. For example, dust deposition during extreme high winds can be 10 to 25 times higher than the annual average (Liu et al., 2004; Zhang et al., 2010; Goudie, 2014). The observations on dust deposition are ordinarily scattered and discontinuous. The limited observational data restricted our understanding of dust fluxes between the atmosphere and land surface, thus numerical simulations are needed to evaluate dust fluxes and the rate of global dust deposition. Ginoux et al. (2001) simulated dust deposition at 16 sites around the world and predicted the annual global dust deposition was approximately 1842 megatons. Shao et al. (2011) estimated that over 2000 megatons of dust is emitted from the Earth's surface into the atmosphere annually. Zheng et al. (2016) estimated that annual average global dust deposition was approximately 1161 megatons. However, uncertainties remain in estimating the dust deposition budget of the earth system because of the lack of observational data and inaccuracies of parameters in numerical simulations (Ginoux et al., 2001; Huneeus et al., 2011; Shao et al., 2011; Ginoux, et al., 2012; Zhang et al., 2014a). Observation of world-wide dust deposition is urgently needed to assess biogeochemical cycle of dust on Earth.

Located in east Asia and at the boundary of central Asia, Xinjiang Province of northwestern China has long played a strategic role in cultural and economic trade between Asia and Europe. Xinjiang Province experiences severe sand and dust storms and is highly susceptible to desertification (Chen, 2010). Xinjiang Province is one of two major source regions of

atmospheric dust in China, the other region being western Inner Mongolia (Xuan, 1999; Xuan et al., 2000). Long-range transport of dust from the region strongly affects air quality in east Asia (Derbyshire et al., 1998; Uno et al., 2009). In fact, dust from the region can be transported across the Pacific Ocean and thus impact air quality in North America (Husar et al., 2001; Osada et al., 2014). Indeed, particulate matter associated with dust transport can severely deteriorate air quality (Sharratt and Lauer, 2006; Shoemaker et al., 2013). Over the past decades, many observations have been made of processes that govern dust emissions, transport, and deposition in Asia (Shao et al., 2011; Groll et al., 2013). Little is known, however, concerning dust deposition and concentrations in Xinjiang Province. In fact, temporal and spatial variations in dust deposition and concentration have not been characterized despite the importance of dust transport from the region. To improve our understanding of the fate and transport of airborne dust in central and east Asia, there is a need for continuous and long-term records of dust deposition and concentration. The purpose of this study is to characterize the spatiotemporal distribution of dust deposition and particulate matter concentration in Xinjiang Province. This characterization will strengthen our comprehension on aerosol transport in east Asia and provide aerosol data for modelling dust transport in global desertification regions.

## 2 Methods

### 2.1 Study area

Xinjiang Province is located in northwest China and is the largest inland province which covers an area of more than 1.6 million $km^2$ (Fig. 1). The Taklamakan and Gurbantunggut Deserts are located in the province. The Taklamakan Desert, located in the southern region, is the world's largest shifting-sand dune desert. The Gurbantunggut Desert, located in the northern region, is the largest fixed-dune desert in China. The province is in part characterized by extreme aridity and aeolian desertification. The average annual precipitation varies from more than 700 mm in high-altitude forests and mountains to less than 50 mm in the deserts. Annual potential evaporation can exceed 2000 mm in desert regions (Chen, 2010). Sand and dust storms occur throughout the year, but are most common in spring. In this study, the province was divided by latitude and longitude into three regions, those being northern Xinjiang, eastern Xinjiang and southern Xinjiang (Table 2).

### 2.2 Experimental Data

Dust deposition and $PM_{10}$ concentration were measured at environmental monitoring stations maintained by the Xinjiang Environmental Protection Administration, a division of the Ministry of Environmental Protection (MEP) in China. Data collected at 14 stations (Fig. 1 and Table 2) were used in this study and represent a spatial distribution within this region. Dust deposition was determined by the gravimetric method and documented at monthly intervals. Glass cylinders were used to monitor dust deposition. Three cylinders (replicates) were installed to monitor dust deposition at each station. The cylinders (0.15 m in diameter, 0.3 m tall, and open at the top) were partly filled with an ethylene glycol ($C_2H_6O_2$) – water

solution prior to deployment. The solution enabled trapping of dust in a liquid media at temperatures below 0℃ and also minimized evaporation from the cylinder. Cylinders were mounted vertically on a tower at approximate 10 m above ground. The mass of dust collected by the cylinders was determined after washing the contents out of the cylinders and oven-drying the contents at 105℃. Dust deposition rate was calculated as the mass of dust per unit area per unit time and expressed in units of t $km^{-2}$ $mon^{-1}$ (MEP, 1994). Monthly and yearly dust deposition data were available through the MEP for the 14 stations from 2000-2013.

Ambient $PM_{10}$ concentration was measured with high volume samplers designed to collect particulate matter by filtration. The samplers were installed at 1.5 m above the ground and equipped with fiberglass filters for trapping $PM_{10}$. $PM_{10}$ concentration was determined based upon gravimetric filter analysis and flow rate of each sampler. Daily $PM_{10}$ concentration data were obtained by the arithmetic mean of four samplers with the sampling time being >18 h for each sampler. $PM_{10}$ was expressed in μg $m^{-3}$ (MEP, 2011). Annual $PM_{10}$ data were available through the MEP for the 14 stations (Xinjiang Statistical Bureau, 2014).

Daily meteorological data including dust days, surface wind speed and precipitation, were collected from the China Meteorological Administration. A dust day was defined by visibility according to World Meteorological Organization protocol; days in which visibility was <10 km at any observation time throughout the day constituted a dust day. The WMO (World Meteorological Organization) further classifies dust days as dust-in-suspension, blowing dust, and dust storms (http://www.wmo.int/pages/prog/www/WMOCodes.html; Shao and Dong, 2006). Dust-in-suspension constitutes days in which dust is emitted at the station at the time of observation and visibility is <10 km, blowing dust constitutes days in which dust or sand is emitted at the station and visibility is 1-10 km, and dust storms constitutes days in which dust or sand is emitted at the station and visibility is <1 km. Observations of visibility and wind characteristics at each station were taken at 3 hour intervals throughout the day.

Daily air pollution index (API) data were obtained from air quality monitoring statistics published by the MEP (http://datacenter.mep.gov.cn). These data were used to illustrate the impact of airborne dust versus other air pollutants on air quality. The API is calculated according to the daily concentration of three main air pollutants (USEPA, 2006; Wang et al., 2013), namely $PM_{10}$, $SO_2$ and $NO_2$. The API is calculated as:

$$API=max(API_i), \tag{1}$$

$$API_i = \frac{API_u - API_L}{C_u - C_L} \times (C_i - C_L) + API_L, \tag{2}$$

Where $API_i$ is the index for pollutant i (i.e., $PM_{10}$, $SO_2$, and $NO_2$), $API_u$ and $API_L$ are the upper and lower limits of the index for a specific category of air quality (i.e. excellent, slight, moderate, moderately severe, and severe), $C_i$ is the observed concentration of pollutant, and $C_u$ and $C_L$ are the upper and lower limits of the pollutant for a specific category of air quality. Information regarding the determination of the API index can be obtained from the MEP (MEP, 2012a, 2012b). Based on the API, air quality was classified as: excellent with an API of 0 to 50, slight pollution with an API of 50 to 100, moderate pollution with an API of 100 to 200, moderately severe pollution with an API of 200 to 300, and severe pollution with an

API of 300 to 500. For the purpose of this study, we used only API data collected in 2010 since annual deposition and $PM_{10}$ concentration appeared to typify that which occurred between 2000 and 2013 in eastern, northern, and southern Xinjiang Province.

Temporal trends in dust deposition, $PM_{10}$ concentration and dust days were evaluated by testing the significance of the slope estimate using a t test at a probability level (P-value) of 0.05.

## 3 Results and discussion

### 3.1 Dust deposition

Detailed information on dust deposition during 2000-2013 was obtained from 14 environmental monitoring stations (Table 2). Annual average dust deposition across all stations in Xinjiang Province was 301.9 t $km^{-2}$. The highest annual deposition occurred in Hotan and Kashgar in southern Xinjiang while the lowest deposition occurred in Karamay in northern Xinjiang. Based upon spatial characteristics in annual dust deposition, deposition increased from north to south across the province (Fig. 2). The annual average dust deposition was 147.8, 194.9 and 663.6 t $km^{-2}$ in northern, eastern and southern Xinjiang, respectively. Generally, the origin of mineral dust could be attributed to both natural and anthropogenic sources (Miller-Schulze et al., 2015). Although dust deposition was relatively low (<150 t $km^{-2}$) for the majority of stations in northern Xinjiang Province, dust deposition was at least 50% higher for stations within the industrial belt on the northern slope of the Tianshan Mountains. This industrial belt includes Changji and Urumqi. High dust deposition in the industrial belt was due to local industry, coal burning and vehicle exhaust (Matinmin and Meixner, 2011; Zhang et al., 2014b). Therefore, the mixing of the anthropogenic aerosol with transported desert dust contributed to deposition in Changji and Urumqi (Li, et al., 2008).

Figures 3 and 4 depicted the temporal variation in dust deposition from 2000-2013. The highest annual deposition occurred in 2012 in southern Xinjiang, 2002 and 2012 in eastern Xinjiang, and 2001 in northern Xinjiang. Over the 14 year period, dust deposition varied with time across Xinjiang Province. The slope estimate of the relation between average dust deposition and year (-6.4±0.1 t $km^{-2}$ $yr^{-1}$) was significant at P=0.05. This trend was most apparent in northern Xinjiang (slope estimate was -5.6±0.1 t $km^{-2}$ $yr^{-1}$) and least apparent in southern Xinjiang (slope estimate was -1.9±0.1 t $km^{-2}$ $yr^{-1}$). High dust deposition occurred in spring in eastern and northern Xinjiang and in spring and summer in southern Xinjiang (Fig. 4). Dust deposition peaked in April in eastern and northern Xinjiang and in May in southern Xinjiang. This corresponds to the onset of wind erosion caused by intensifying zonal flow and rising air temperatures before the arrival of the summer monsoon (Song et al., 2016). The maximum monthly average dust deposition was 97.5 t $km^{-2}$ in southern Xinjiang, which was 6.9 and 8 times more than the deposition in northern and eastern Xinjiang, respectively. These results suggest that dust deposition in south Xinjiang is of similar magnitude to deposition that occurs in the Middle East and Sahel regions (Khalaf and Al-Hashash, 1983; McTainsh and Walker, 1982; O'Hara et al., 2006).

## 3.2 PM$_{10}$ concentration

The annual average PM$_{10}$ concentration in Xinjiang was 125 μg m$^{-3}$ based upon data collected at 14 stations from 2000-2013. Ten stations (71 percent) in our study had an annual average PM$_{10}$ concentration above the People's Republic of China Class II residential standard of 70 μg m$^{-3}$. The highest annual average PM$_{10}$ concentration (352 μg m$^{-3}$) occurred in Hotan in southern Xinjiang while the lowest average PM$_{10}$ concentration (46 μg m$^{-3}$) occurred in Tacheng in northern Xinjiang. The annual average PM$_{10}$ concentration appeared to increase from northern to southern regions (Fig. 5). Annual average PM$_{10}$ concentration in Xinjiang ranged from 100 to 196 μg m$^{-3}$ (Fig. 6) across years. The annual average PM$_{10}$ concentration was 70, 115 and 239 μg m$^{-3}$ in northern, eastern and southern Xinjiang, respectively. The high annual concentration in southern Xinjiang is of the same magnitude as found in other desertification regions of the world such as South Asia, Middle East, and western Sahel desert (WHO, 2015). These high concentrations of suspended particulates, especially finer particulate, may influence the health of sensitive populations who are susceptible to respiratory illness (Goudie, 2014).

Over the period of record (2000-2013), there was a trend for decreasing PM$_{10}$ concentration in Xinjiang Province. The slope estimate of the relation between annual PM$_{10}$ concentration and year (-4.2±0.1 μg m$^{-3}$ yr$^{-1}$) was significant at P=0.05. This trend was most apparent in southern Xinjiang (slope estimate was -11.8±0.1 μg m$^{-3}$ yr$^{-1}$). On the contrary, PM$_{10}$ concentration appeared to increase with time in eastern and northern Xinjiang (slope estimates were 1.3±0.1 μg m$^{-3}$ yr$^{-1}$ and 3±0.1 μg m$^{-3}$ yr$^{-1}$, respectively). The slope estimate, however, was not statistically different from zero and indicated no apparent trend in PM$_{10}$ concentration with time in northern Xinjiang.  A decrease in both dust deposition and PM$_{10}$ concentration over 2000 to 2013 suggests a positive relationship between dust deposition and PM$_{10}$ concentration. This relationship is supported by data in Fig. 7. Dust particles are delivered back to the surface by both dry and wet deposition (Shao, 2000). In arid and semi-arid region of central Asia, the deposition process is mainly dominated by dry deposition because of less precipitation, which is comprised of gravitational settling, turbulent diffusion and molecular diffusion (Zhang and Shao, 2014; Xi and Sokolik, 2015). Those physical processes from the air to surface are complex and dependent on dust concentration with a representation of the higher the dust concentration, the higher the dust deposition (Slin and Slin, 1980; Wesely and Hicks, 2000; Petroff, et al., 2008; Zhang et al., 2014a). Fig. 7 showed that dust deposition significantly increased with high PM$_{10}$ concentration above 200 μg m$^{-3}$. A logarithmic function fit the data better than a linear function, suggesting that changes in atmospheric PM$_{10}$ concentration are smaller at higher rates of deposition with a correlation coefficient R$^2$≥0.81. This trend could be due to deposition of larger or more massive particles under more severe dust or sand storms. While PM$_{10}$ concentration may rise under more severe wind erosion events, the limited supply of PM$_{10}$ in sand (major soil type in the province) will likely suppress a rise in PM$_{10}$ concentration in the atmosphere under more severe erosion events. Nevertheless, from 2000 to 2013, the decline in both dust deposition and PM$_{10}$ concentration across Xinjiang could be due to less frequent or intense dust storms because dust deposition in major cities of northern China was found to be closely related to the frequency of sand and dust storms (Zhang et al., 2010).

### 3.3 Influence of atmospheric dust deposition on local air quality

Daily ambient air quality has been reported by MEP since 2000. Airborne dust is one of three pollutants influencing the API, thus the relative contribution of dust to the API was of interest. Accordingly, we made a comparative analysis to identify the impact of airborne dust on air quality in Urumqi and Kuytun in northern Xinjiang, Turpan and Hami in eastern Xinjiang, and

Kashgar and Hotan in southern Xinjiang (Fig. 8). In 2010, there were 178, 286, 351, 334, 363, and 360 days in which $PM_{10}$ was the main constituent of the API in Kuytun, Urumqi, Turpan, Hami, Kashgar and Hotan, respectively (Fig. 8). The $PM_{10}$ constituent accounted for 48.7%, 78.4%, 96.2%, 91.5%, 99.5%, and 99.6% of the API in the respective above cities. These data suggest that particulate matter is the main air pollutant in Xinjiang. Severe $PM_{10}$ pollution (API >300) occurred mainly in spring, which was closely associated with the seasonality of strong winds and dust storm activity (Li et al., 2004). Stations

in southern Xinjiang (Kashgar and Hotan) had higher API's caused by elevated $PM_{10}$ concentrations throughout the year. This can be attributed to the violent and persistent eaolian activity around the Taklamakan Desert (Pi et al., 2014). Consequently, $PM_{10}$ is an important pollutant which dominates ambient air quality in Xinjiang.

### 3.4 Factors influencing dust deposition and $PM_{10}$ concentration

Many factors influence ambient particulate concentration and dust deposition, but weather appears to be a dominate factor in

arid regions (Zhang et al., 1996; Zhang et al., 2010). In fact, dust activity is highly correlated with variability in global climate and atmospheric circulation (Gong, et al, 2006; Mao et al., 2011; Shao et al., 2013). The Eurasian atmospheric circulation greatly affects weather in central and east Asia (Zhang et al., 1997; Kang, et al., 2013; Xi and Sokolik, 2015). Dust activities are primarily driven by the strength of cyclone and the Siberian High affecting the study region (Park, et al., 2011; Shao et al., 2013). Strong winds associated with this atmospheric circulation cause large amounts of dust to be emitted

into the atmosphere. Deserts in central Asia are an important source of atmospheric mineral dust (Miller-Schulze et al., 2015). Under the strong westerly circulation, atmospheric dust can be transported a few hundred kilometers to the east and be deposited through wet scavenging and dry settling (Shao, 2000; Chen et al., 2014). Despite the Taklimakan and Gurbantunggut Deserts being local sources of dust in Xinjiang Province, long-range transport of dust from the central Asian Aralkum, Karakum, Caspian and Kyzylkum Deserts (Indoitu, et al., 2012) could also contribute to the dust deposition and

ambient $PM_{10}$ concentration in neighbouring Xinjiang Province. Since the 1980s, the Aralkum Desert in Uzbekistan and Kazakhstan has become one of world's youngest deserts and a potential source of salt dust in east Asia (Indoitu, et al., 2012; Groll, et al., 2013; Opp, et al., 2016).

Climate also directly influences the atmospheric environment of arid and semi-arid areas (Wei et al., 2004; Zu et al., 2008; Huang et al., 2014). The annual average precipitation in north, east and south Xinjiang is 237, 94, and 87 mm, respectively.

Dust emission was negatively correlated with precipitation (Gong, et al., 2006). Therefore, the lack of precipitation contributes to dust emissions. In fact, regions with lower precipitation have higher $PM_{10}$ concentrations and dust deposition in Xinjiang Province. Daily average wind speed in north, east and south Xinjiang is 2.5, 2.2, and 1.8 m s$^{-1}$, respectively. In

contrast to precipitation, regional differences in wind speed fail to account for differences in $PM_{10}$ concentrations and dust deposition. Dust distribution in south Xinjiang (including the Tarim Basin and Taklamakan Desert), however, is strongly affected by wind flow patterns. Aeolian transport in the Taklamakan Desert is predominantly from the northeast toward the south (Wang et al., 2014; Rittner et al., 2016). Huang et al. (2014) reported that the Taklamakan Desert is a source of fine dust particles (≤3μm in aerodynamic diameter) which significantly influences East Asia. Strong northeast winds dominate the prevailing wind regime in the eastern Taklamakan Desert; these winds influence air quality in both the eastern and the southeastern parts of the desert. The western and northern parts of the Taklamakan Desert and Tarim Basin are highly affected by west, northwest and north winds (Sun and Liu, 2006; Zan et al., 2014; Li et al., 2015). Under prevailing winds, dust aerosols are transported from northern to the southern Taklamakan Desert (e.g. Hotan city) and thereby cause high ambient $PM_{10}$ concentration and dust deposition.

Spatial differences in dust deposition and $PM_{10}$ concentration across Xinjiang Province may also be due in part to differences in frequency of dust days in the region. Dust storms normally occurred in all seasons in southern Xinjiang. The magnitude of wind erosion and dust day frequency in southern Xinjiang is nearly twice as large as in northern and the eastern Xinjiang (Wang et al., 2006). Figure 9 displays the variation in dust day frequency in Xinjiang Province from 2000 to 2013. The data indicate that the annual average frequency of dust days fluctuated from 15 to 52 days. The frequency of dust days in the southern region ranged from 41 to 133 days while the frequency of dust days in eastern and northern regions ranged from 2-45 days and 0-3 days, respectively, across years. The slope estimate of the relationship between dust days and years (0.11 day $yr^{-1}$) indicated no apparent trend for an increase or decrease in dust days from 2000 to 2013. Thus, despite no temporal trend in dust days, we observed a decline in dust deposition and $PM_{10}$ concentration across years. This decline in dust deposition or $PM_{10}$ concentration could be due to a decrease in frequency of severe dust days versus frequency of dust days from 2000 to 2013 in the region. We are unaware of any previous study which has examined dust day severity in Xinjiang Province, thus we used data available through the China Meteorological Administration to assess trends in dust day severity. Dust days were characterized according to dust-in-suspension, blowing dust, and dust storm events. Although there was no trend in the frequency of blowing dust and dust storm events from 2000 to 2013, there was a trend for fewer dust-in-suspension events from 2000 to 2013 (Fig. S1 in the Supplement). Thus, there appeared to be a close association between frequency of dust-in-suspension events and $PM_{10}$ concentration and dust deposition. Nevertheless, in examining the relationship between average annual dust days and dust deposition or $PM_{10}$ concentration across stations, the frequency of dust days was closely related to dust deposition ($R^2=0.93$) (Fig.10) and ambient $PM_{10}$ concentration ($R^2=0.89$) (Fig.11). There was a significant increase in dust deposition (7.91 t $km^{-2}$ $day^{-1}$) and $PM_{10}$ concentration (2.06 μg $m^{-3}$ $day^{-1}$) associated with an increase in dust days.

**4 Conclusions**

The atmospheric environment of central and east Asia is severely affected by the airborne dust, thus this study was undertaken to quantify dust deposition and ambient $PM_{10}$ concentration in east Asia. Data collected at 14 environmental monitoring stations from 2000-2013 in Xinjiang Province, China confirmed that annual average dust deposition ranged from 255.7 to 421.4 t $km^{-2}$. Annual average $PM_{10}$ concentration varied from 100 to 196 μg $m^{-3}$. The highest dust deposition was observed in Hotan in the southern Taklamakan Desert with 1394.1 t $km^{-2}$, which is ten times that in China's Loess Plateau (Liu et al., 2004). The highest ambient $PM_{10}$ concentration was also observed in Hotan with 352 μg $m^{-3}$, which far exceeds the World Health Organization's long-term exposure standard (WHO, 2014). These observation results provide a concrete evidence on the study area as "dust region" described by Shao et al. (2011) and Ginoux et al. (2012), and suggest that dust source in east Asia affect regional air quality and is a potential contributor of global dust.

The spatial distribution and temporal variability in dust deposition and ambient $PM_{10}$ concentration showed significant variation and a trend for lower deposition and concentration with time. The inner-annual dynamic of dust deposition varied significantly with seasonality. Spring and summer had the highest dust deposition (1.3 times the average), followed by autumn and winter. The highest intensity of dust deposition was observed in May, followed by April, June and July.

In dust source areas such as Xinjiang, China, windblown sand and dust affect air quality, especially during the spring season. The analysis of the data indicated no trend in frequency of dust days from 2000 to 2013. A positive relationship existed, however, between dust days and dust deposition as well as airborne $PM_{10}$ concentration across stations. The effect of weather on dust deposition and ambient air quality cannot be expressed by a simple correlation and should not be extrapolated based on the currently limited evidence. This study provides information on the potential spatial-temporal dust deposition and ambient dust aerosol variation in east Asia. Although longer term datasets are needed to address trends over longer time periods, this work can aid in adjusting model parameters in simulating dry dust deposition or $PM_{10}$ concentration in desertification regions of east Asia.

*Acknowledgements.* The authors would like to thank anonymous reviewers for their useful comments that contributed to improve the manuscript. This work was supported by the National Natural Science Foundation of China (No.41301655), the West Light Foundation of the Chinese Academy of Sciences (No.XBBS201104), and the Open Funds (No.LAPC-KF-2013-17) of the State Key Laboratory of Atmospheric Boundary Layer and Atmospheric Chemistry, China.

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

**Tables**

**Table 1.** Observations of dust deposition in desertification regions.

| Continent | Location | Period | Dust deposition t km$^{-2}$ yr$^{-1}$ | Citation |
|---|---|---|---|---|
| America | Kansas, USA | 1964-1966 | 53.5-62.1 | USDA (1968) |
| | New Mexico, USA | 1962-1972 | 9.3-125.8 | Gile and Grossman (1979) |
| | Arizona, USA | 1972-1973 | 54 | Péwé (1981) |
| Europe | Spain | 2002-2003 | 17-79 | Menéndez et al. (2007) |
| Africa | Nigeria | 1976-1979 | 137-181 | McTainsh and Walker (1982) |
| | Niger | 1985 | 164-212 | Drees et al. (1993) |
| | Libya | 2000-2001 | 420 | O'Hara et al. (2006) |
| Oceania | Australia | 2000-2001 | 5-100 | Cattle et al. (2009) |
| Asia | Israel | 1968-1973 | 57-217 | Yaalon and Ganor (1975) |
| | Kuwait | 1982 | 2600 | Khalaf and Al-Hashash (1983) |
| | Saudi Arabia | 1991-1992 | 4704 | Modaihash (1997) |
| | Lanzhou, China | 1988-1991 | 108 | Derbyshire et al. (1998) |
| | Loess Plateau, China | 2003-2004 | 133 | Liu et al. (2004) |
| | Urumqi, China | 1981-2004 | 284.5 | Zhang et al. (2010) |
| | Iran | 2008-2009 | 72-120 | Saeid et al. (2012) |
| | Uzbekistan | 2003-2010 | 8365 | Groll et al. (2013) |

**Table 2.** Dust deposition and $PM_{10}$ concentrations at 14 stations in Xinjiang.

| No. | Station | Region* | Latitude | Longitude | Population** (million) | Annual dust deposition (t km⁻²) | Annual $PM_{10}$ concentration (μg m⁻³) |
|---|---|---|---|---|---|---|---|
| 1 | Urumqi | NJ | 43.832 °N | 87.616 °E | 2.26 | 229.4 | 141 |
| 2 | Changji | NJ | 44.017 °N | 87.308 °E | 0.36 | 295.7 | 76 |
| 3 | Shihezi | NJ | 44.306 °N | 86.080 °E | 0.62 | 107.7 | 61 |
| 4 | Bole | NJ | 44.900 °N | 82.071 °E | 0.27 | 133 | 48 |
| 5 | Karamay | NJ | 45.580 °N | 84.889 °E | 0.29 | 81.1 | 54 |
| 6 | Tacheng | NJ | 46.691 °N | 82.952 °E | 0.17 | 84.9 | 39 |
| 7 | Yining | NJ | 43.912 °N | 81.329 °E | 0.53 | 142.7 | 78 |
| 8 | Kuytun | NJ | 44.426 °N | 84.903 °E | 0.30 | 108.1 | 66 |
| 9 | Hami | EJ | 42.818 °N | 93.514 °E | 0.48 | 209.8 | 84 |
| 10 | Turpan | EJ | 42.957 °N | 89.179 °E | 0.28 | 180.1 | 145 |
| 11 | Korla | SJ | 41.727 °N | 86.174 °E | 0.57 | 231.8 | 131 |
| 12 | Hotan | SJ | 37.113 °N | 79.922 °E | 0.33 | 1394.1 | 352 |
| 13 | Kashgar | SJ | 39.471 °N | 75.989 °E | 0.57 | 516.9 | 236 |
| 14 | Aksu | SJ | 41.170 °N | 80.230 °E | 0.51 | 511.5 | 238 |

* Xinjiang Province was classified into three regions: northern Xinjiang (NJ), eastern Xinjiang (EJ), and southern Xinjiang (SJ).

**Population in 2013 as reported by the Xinjiang Statistical Bureau.

**Figures**

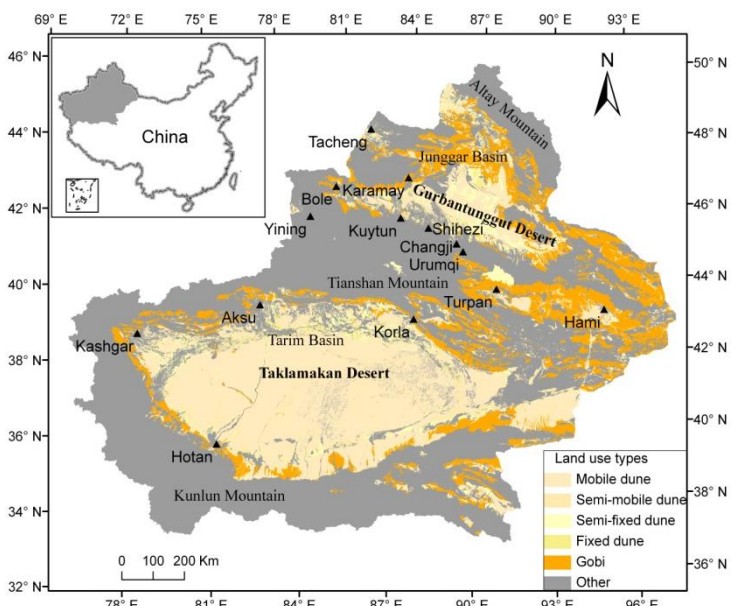

**Figure 1.** Location of Xinjiang Province in China (gray area outlined on inset map). Dust deposition and concentrations were measured at stations signified by small triangles. Land use types are identified across the province according to Wang et al., 2005.

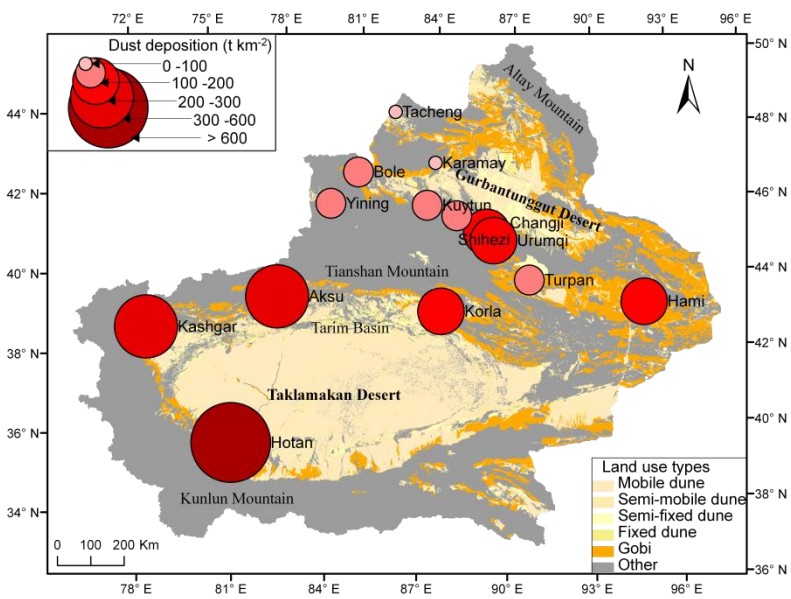

**Figure 2.** Annual average dust deposition reported at 14 stations in Xinjiang Province from 2000-2013. Land use types across the province are identified according to Wang et al. (2005).

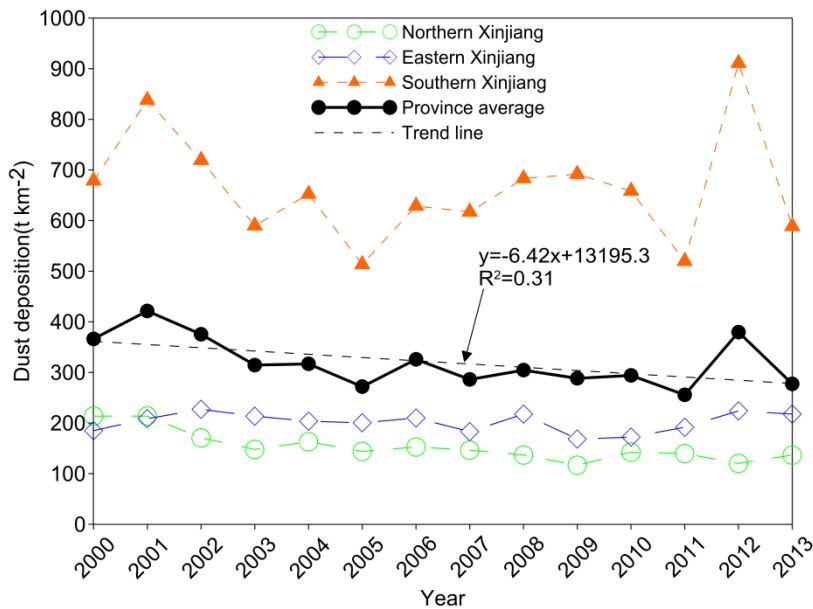

**Figure 3.** Annual average dust deposition in Xinjiang Province from 2000 to 2013. Dust deposition in northern, eastern and southern Xinjiang is the average deposition at 8, 2 and 4 stations, respectively.

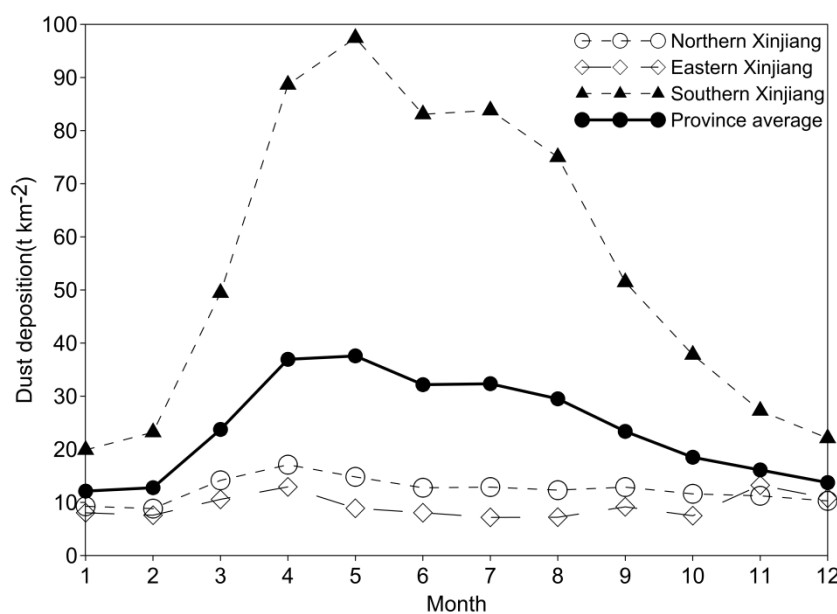

**Figure 4.** Monthly average dust deposition in Xinjiang Province from 2000-2013. Dust deposition in northern, eastern and southern Xinjiang is the average deposition at 8, 2 and 4 stations, respectively.

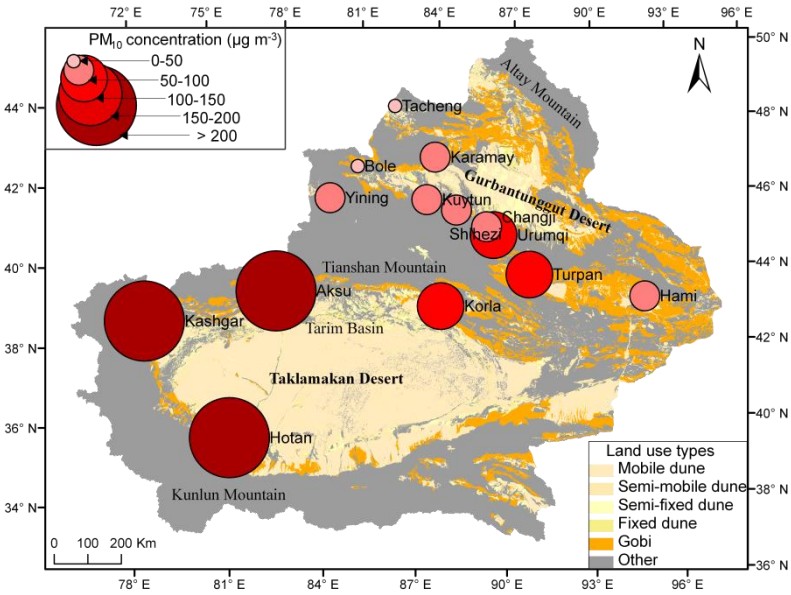

**Figure 5.** Annual average $PM_{10}$ concentration reported for 14 stations in Xinjiang Province from 2000-2013. Land use types are identified across the province according to Wang et al., 2005.

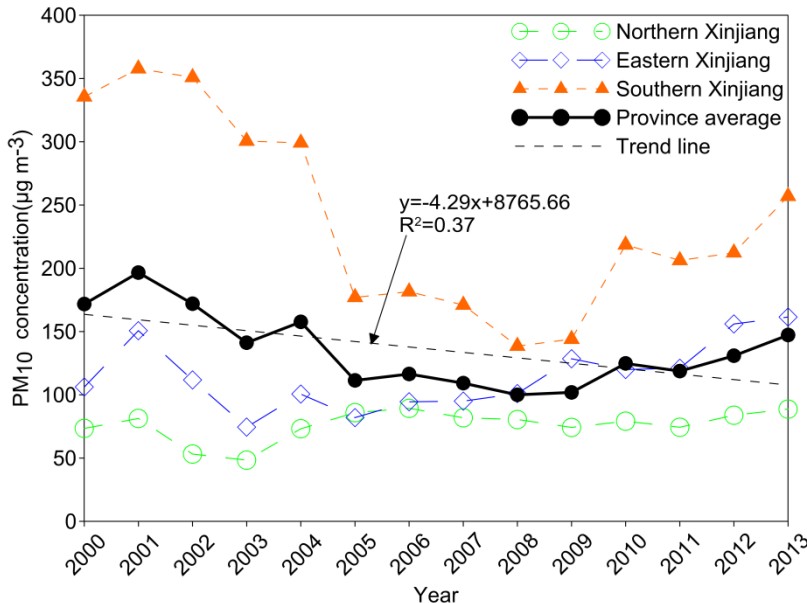

**Figure 6.** Annual average PM$_{10}$ concentration in Xinjiang Province from 2000 to 2013.

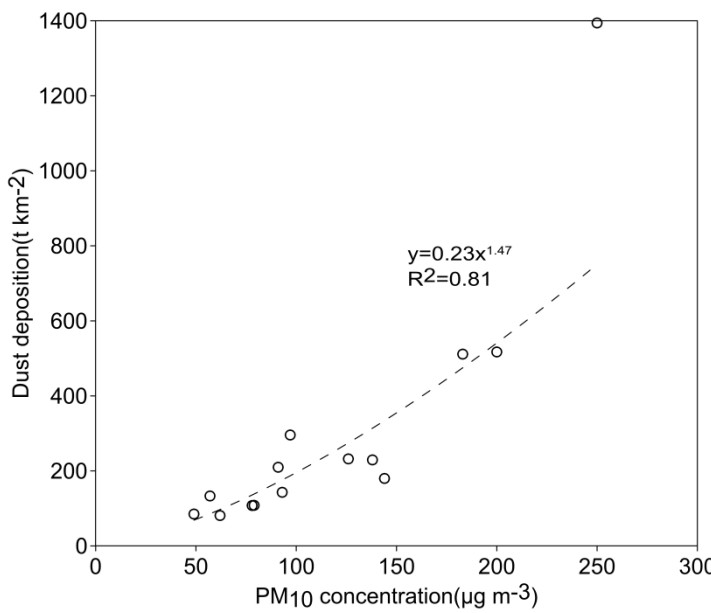

**Figure 7.** Relationship between annual dust deposition and $PM_{10}$ concentration in Xinjiang Province. Each point represents data averaged across 2000 to 2013 at one station.

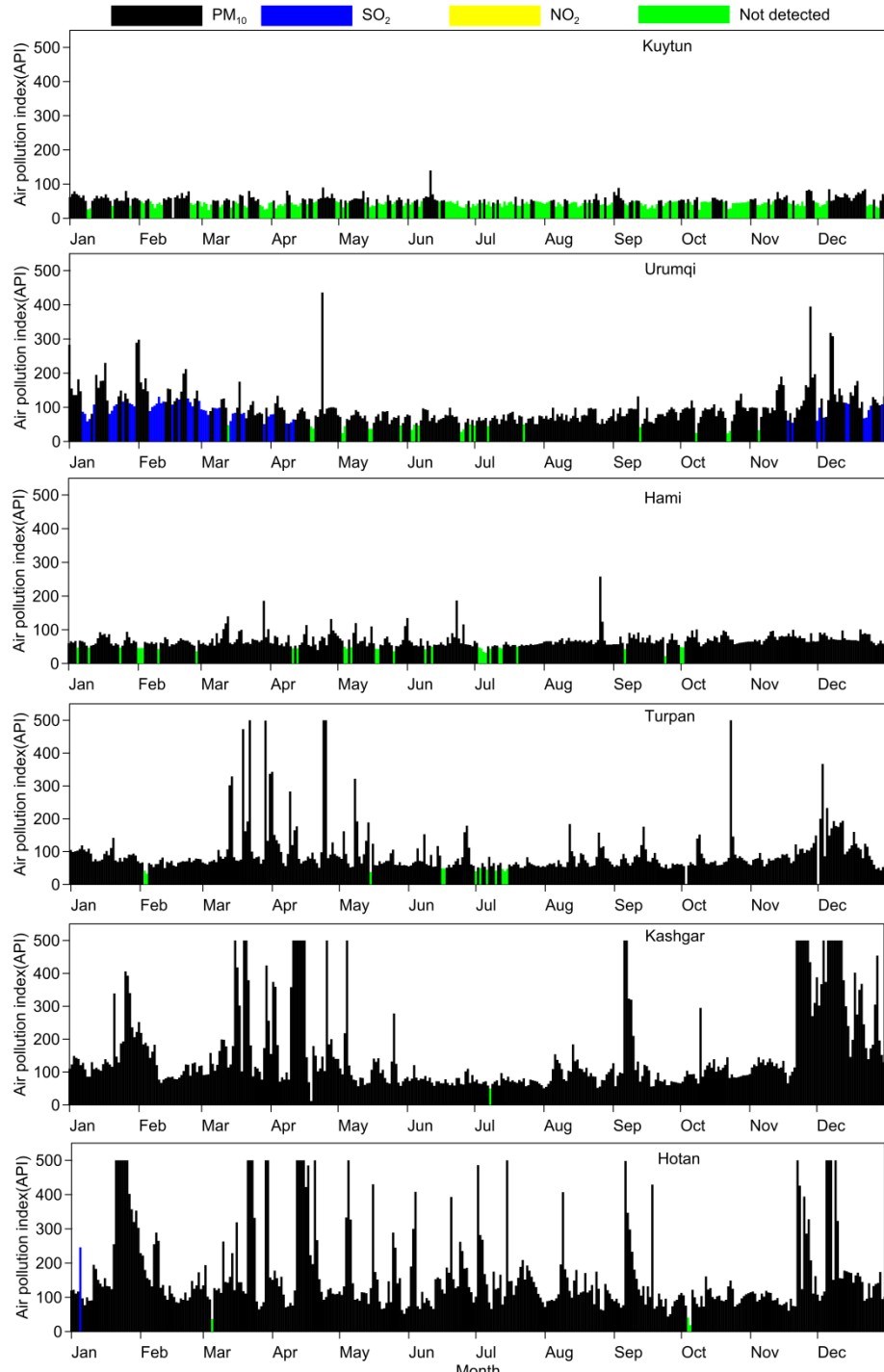

**Figure 8.** Daily air pollution index for Kuytun and Urumqi in northern Xinjiang, Hami and Turpan in eastern Xinjiang, and Kashgar and Hotan in southern Xinjiang in 2010. The main air pollutant contributing to the daily API is identified for each station. Not detected indicates excellent air quality (API<50).

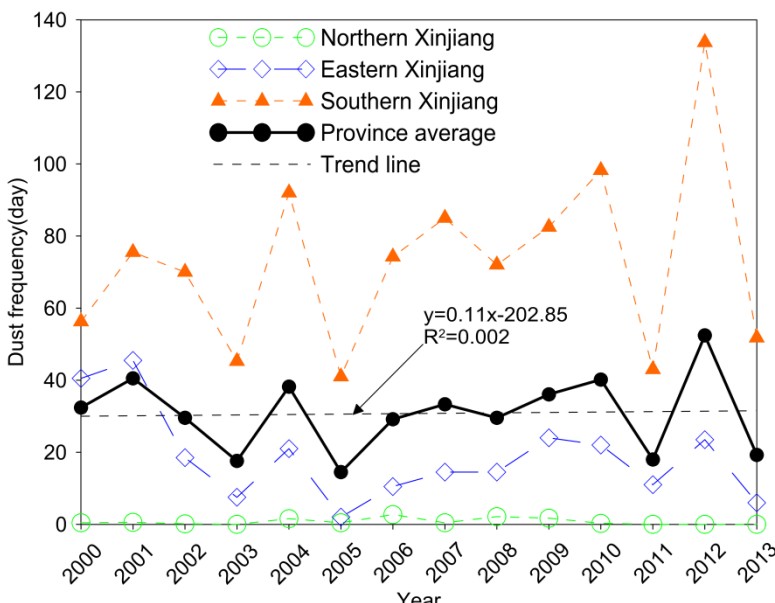

**Figure 9.** Dust day frequency in Xinjiang Province from 2000 to 2013.

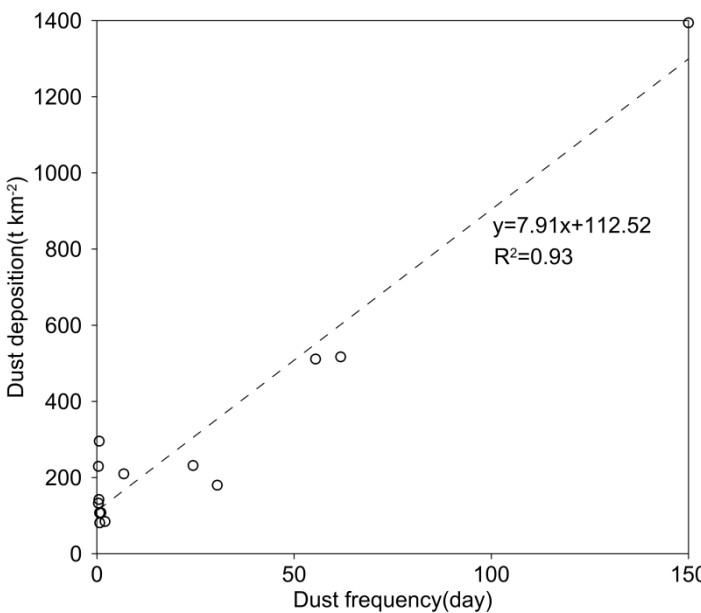

**Figure 10.** Relationship between annual dust deposition and dust day frequency in Xinjiang Province. Each point represents data averaged across 2000 to 2013 at one station.

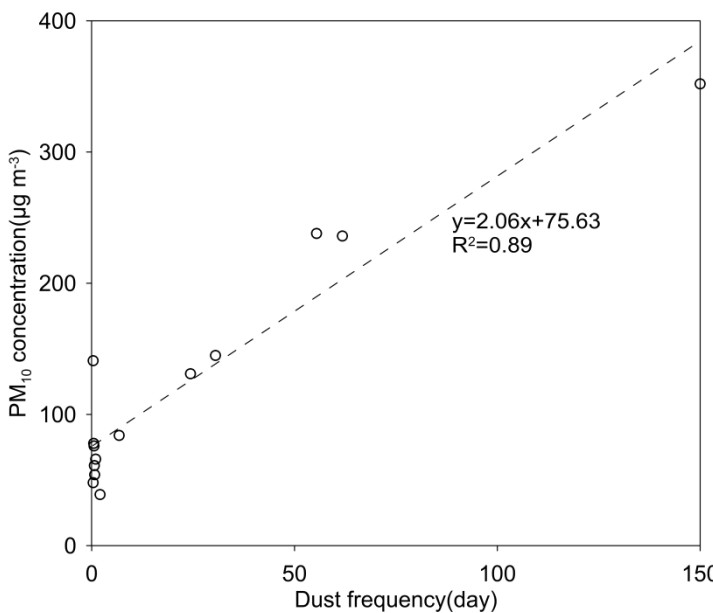

**Figure 11.** Relationship between $PM_{10}$ concentration and dust day frequency in Xinjiang Province. Each point represents data averaged across 2000 to 2013 at one station.