# Peer review of "Dust deposition and ambient PM10 concentration in northwest China: Spatial and temporal variability"

_Atmospheric Chemistry and Physics, 2016_

## Referee Comment (RC1) · Anonymous Referee #1 · 21 Jul 2016

This research paper presents an analysis of spatial and temporal characteristic of the dust deposition and ambient PM10 concentration in Xinjiang Province of China. The manuscript thus constitutes some original contribution to the understanding dust transport and modeling in this region, where dust deposition data is scanted. This paper will be of interest to readers of ACP.

Recommendation Accept after major revision. In general, I support the potential publication of this paper due to its scientific interest. However, the manuscript needs to be extensively improved in various aspects, otherwise it would be difficult to support its final publication. I strongly advice the authors to take into consideration of the following major and minor remarks so as to improve the quality this manuscript.

Major comments

1. The title of the paper is "Dust deposition and ambient PM10 concentration in central Asia: Spatial and temporal variability", while the paper only focus on Xinjiang province, China. However, the Central Asia is generally referred to the core region of the Asian continent which usually including Kazakhstan, Kyrgyzstan, Tajikistan, Turkmenistan, and Uzbekistan in the modern context. The title of the paper, therefore, should be reconsidered regarding study region and avoid possible confusion.

2. In the Introduction, the author cites Simonson (1995) and Pye(1987) to show that dust plays an important role in climate change and environmental quality. The paper cited here, which is published in 1990s and relatively outdated. Since then, the dust deposition data have been enriched, as listed in the Table.1. What is the current circumstance of dust deposition research? What is still unknown? I would suggest author include more recent paper in this field to strengthen the introduction section. The following paper is for reference only.

(1)Shao, Y., et al., 2011: Dust cycle: An emerging core theme in Earth system science. Aeolian Research, 2.4 (2011): 181-204.

(2)Chen, S., Huang, J., Zhao, C., Qian, Y., Leung, L. R., and Yang, B.: Modeling the transport and radiative forcing of Taklimakan dust over the Tibetan Plateau: A case study in the summer of 2006, J. Geophys. Res. Atmos., 118, 797–812.

(3)Chen, S., Zhao, C., Qian, Y., Leung, L. R., Huang, J., Huang, Z., Bi, J., Zhang, W., Shi, J., Yang, L., Li, D., and Li, J.: Regional modeling of dust mass balance and radiative forcing over East Asia using WRF-Chem, Aeolian Res., 15, 15–30.

(4)Huang, J., T. Wang, W. Wang, Z. Li, and H. Yan, 2014: Climate effects of dust aerosols over East Asian arid and semiarid regions, Journal of Geophysical Research: Atmospheres, 119, 11398–11416, doi:10.1002/2014JD021796.

(5)Huang, J., Fu, Q., Su, J., Tang, Q., Minnis, P., Hu, Y., Yi, Y., and Zhao, Q.: Taklimakan dust aerosol radiative heating derived from CALIPSO observations using the Fu-Liou radiation model with CERES constraints, Atmos. Chem. Phys., 9, 4011–4021, doi:10.5194/acp-9-4011-2009, 2009.

3. The paper did not provide any discussion regarding dust source for deposition and PM10 in this region and thus the analysis was based on the unstated assumption that the only two dust sources are Taklimakan desert and Gurbantunggut desert. However, this might not be the case all the time, since long-range transport of dust from central Asia could also contribute to the dust deposition in Xinjiang province, despite the two large local dust sources. Without the analysis of dust source in the first place, the attempt to explain the spatial and temporal characteristic of dust deposition and ambient PM10 seems unwarranted. I would recommend the authors give a brief discussion of dust sources in the revised manuscript.

4. In this study only one factor are considered and examined in Section 3.4, which is dust days, while the subtitle of the section mentioned "factors." Although dust event might be of the dominant factor, other factors should also be taken in account or at least be mentioned in the analysis. For instance, it is widely recognized that the wind speed and direction could be very influential to dust transport. In the manuscript, although data of wind speed and direction is mentioned in the section 2.2(Line 31), the analysis regarding this data was not provided in the manuscript. In addition to the wind, precipitation could also be a controlling factor. Further analysis of the other factors should also be provided in the manuscript.

Minor comments

Page 1, Line 17 ...(particulate matter $\leq$10 $\mu$m in aerodynamic diameter)... Please rephrase the sentence in the parentheses.

Page 1, Line 26-27 ...The arid climate likely influenced the high dust deposition and PM10 concentration in the region... This sentence is uncorroborated by the manuscript.

Page 1, Line 29 This study suggests that sand storms are a major factor affecting... Please change "are" to "is".

Page 2, Line 7-8 An understanding of atmospheric dust sources, emissions, and deposition is therefore essential to improve regional air quality. This sentence is not logically related to the information given before it. The discussion prior to it cannot lead to the conclusion that this kind understanding can be helpful to improve regional air quality.

Page 2, Line 30 ... that spans the 21st century. The sentence is overstated, since only 2000-2013 was analyzed in the study, which certainly did not span 21st century.

Page 3, Line 31 Daily meteorological data, including surface wind speed and direction ... Even though the surface wind speed and direction are mentioned in the data description, the analyses relating to them are not given in the manuscript.

Page 5, Line 2 This industrial belt includes Changji and Urumqi. High dust deposition in the industrial belt was due to industry, coal burning and vehicle exhaust. This explanation is possible, with the anthropogenic source of dust is considered. Please further strengthen this conjecture with relevant papers. In addition to the Changji and Urumqi, Hami, which is also a city located in northern Xinjiang, also had a high dust deposition value as depicted in Figure.2. Why?

Page 5, Line 14 .. data suggest that particulate matter is the main air pollutant in Xinjiang. The PM10 constituent accounted for 48.7% and 48.2% of the API in the Kuytun and Urumqi. It is necessarily suggest the particulate matter is the main air pollutant?

Page 6, Line 31—Page 7, Line 1-11 This decline in dust deposition or PM10 concentration could be due to a decrease in frequency of severe dust days versus frequency of dust days from 2000 to 2013 in the region.... Nevertheless, in examining the relationship between average annual dust days and dust deposition or PM10 concentration across stations, the frequency of dust days was closely related to dust deposition

(R2=0.93) (Fig.10) and ambient PM10 concentration (R2=0.89) (Fig.11). There was a significant 10 increase in dust deposition (7.91 t km-2 day-1) and PM10 concentration (2.06 $\mu$g m-3 day-1) associated with an increase in dust days. In this paragraph, the relationship between dust deposition/PM10 concentration and dust day frequency at each station is investigated. The result, admittedly, is evident show there is a connection. According to the definition of different dust days, which can be found in section 2.2(page4,line1-5), blowing dust and dust storm constitutes days in which dust is emitted at the station, while dust-in-suspension constitutes days in which dust is not emitted at the station. However, the scatter plot fails to distinguish the inherent difference between these three dust events. Moreover, since the dust is not emitted at this station during dust-in-suspension days, the conclusion given by author, that there appeared to be a close association between frequency of dust-in-suspension events and dust deposition, become unconvincing.

Page 10, Figure 2 Please add units for dust deposition in the legend within the figure.

Page 17, Figure 5 Please add units for PM10 concentration in the legend within the figure.

---

## Author Comment (AC1) · 29 Jul 2016

We thank anonymous referee #1 for his/her supportive and thoughtful remarks.

Major comments

Question 1: The title of the paper is "Dust deposition and ambient PM10 concentration in central Asia: Spatial and temporal variability", while the paper only focus on Xinjiang province, China. However, the Central Asia is generally referred to the core region of the Asian continent which usually including Kazakhstan, Kyrgyzstan, Tajikistan, Turkmenistan, and Uzbekistan in the modern context. The title of the paper, therefore, should be reconsidered regarding study region and avoid possible confusion.

[Figure]

Reply: We have revised the paper to indicate that Xinjiang Province is located in east Asia. While Xinjiang Province is technically located at the western fringe of east Asia, the geography of the Province is very similar to central Asia. The title has been changed to "Dust deposition and ambient PM10 concentration in northwest China: Spatial and temporal variability".

Question 2: In the Introduction, the author cites Simonson (1995) and Pye (1987) to show that dust plays an important role in climate change and environmental quality. The paper cited here, which is published in 1990s and relatively outdated. Since then, the dust deposition data have been enriched, as listed in the Table.1. What is the current circumstance of dust deposition research? What is still unknown? I would suggest author include more recent paper in this field to strengthen the introduction section.

Reply: New citations have been added to the Introduction to support the discussion of the importance of dust in climate change and environmental quality. Citations added to the Introduction include:

Chen, S., Huang, J., Zhao, C., Qian, Y., Leung, L. R., and Yang, B.: Modeling the transport and radiative forcing of Taklimakan dust over the Tibetan Plateau: A case study in the summer of 2006, J. Geophys. Res. Atmos., 118, 797–812.

Chen, S., Zhao, C., Qian, Y., Leung, L. R., Huang, J., Huang, Z., Bi, J., Zhang, W., Shi, J., Yang, L., Li, D., and Li, J.: Regional modeling of dust mass balance and radiative forcing over East Asia using WRF-Chem, Aeolian Res., 15, 15–30.

Huang, J., T. Wang, W. Wang, Z. Li, and H. Yan, 2014: Climate effects of dust aerosols over East Asian arid and semiarid regions, Journal of Geophysical Research: Atmospheres, 119, 11398–11416, doi:10.1002/2014JD021796.

Huang, J., Fu, Q., Su, J., Tang, Q., Minnis, P., Hu, Y., Yi, Y., and Zhao, Q.: Taklimakan dust aerosol radiative heating derived from CALIPSO observations using the

Fu-Liou radiation model with CERES constraints, Atmos. Chem. Phys., 9, 4011–4021, doi:10.5194/acp-9-4011-2009, 2009.

Lin, Y., and Feng, J.: Aeolian dust contribution to the formation of alpine soils at Amdo (Northern Tibetan Plateau), Geoderma, 259-260, 104-115, doi:10.1016/j.geoderma.2015.05.012, 2015.

Shao Y P.: Physics and modeling of wind erosion, Dordrecht, Kluwer Academic Publishers, 225-278, 2000.

Varga, G., Cserháti, C., Kovács, J., and Szaliai, Z.: Saharan dust deposition in the Carpathian Basin and its possible effects on interglacial soil formation, Aeolian Research, 22, 1-12, doi:10.1016/j.aeolia.2016.05.004, 2016.

Zheng, Y., Zhao, T., Che, H., Liu, Y., Han, Y., Liu, C., Xiong, J., Liu, J., and Zhou, Y.: A 20-year simulated climatology of global dust aerosol deposition, Science of The Total Environment, 557-558, 861-868, doi:10.1016/j.scitotenv.2016.03.086 , 2016.

Question 3: The paper did not provide any discussion regarding dust source for deposition and PM10 in this region and thus the analysis was based on the unstated assumption that the only two dust sources are Taklimakan desert and Gurbantunggut desert. However, this might not be the case all the time, since long-range transport of dust from central Asia could also contribute to the dust deposition in Xinjiang province, despite the two large local dust sources. Without the analysis of dust source in the first place, the attempt to explain the spatial and temporal characteristic of dust deposition and ambient PM10 seems unwarranted. I would recommend the authors give a brief discussion of dust sources in the revised manuscript.

Reply: A brief discussion on dust sources in central and east Asia has been added to Section 3.4 to include the following: Deserts in central Asia are a source of atmospheric mineral dust (Miller-Schulze et al., 2015). Under the strong westerly circulation, atmospheric dust can be transported a few hundred kilometers to the east and

be deposited through wet scavenging and dry settling (Shao, 2000; Chen et al., 2014). Despite the Taklimakan and Gurbantunggut Deserts being local sources of dust in Xinjiang Province, long-range transport of dust from the central Asian Aralkum, Karakum, Caspian and Kyzylkum Deserts (Indoitu, et al., 2012) could also contribute to the dust deposition and ambient PM10 concentration in neighboring Xinjiang Province. Since the 1980s, the Aralkum Desert in Uzbekistan and Kazakhstan has become one of world's youngest deserts and a potential source of salt dust in east Asia (Indoitu, et al., 2012; Groll, et al., 2013; Opp, et al., 2016).

Question 4: In this study only one factor are considered and examined in Section 3.4, which is dust days, while the subtitle of the section mentioned "factors." Although dust event might be of the dominant factor, other factors should also be taken in account or at least be mentioned in the analysis. For instance, it is widely recognized that the wind speed and direction could be very influential to dust transport. In the manuscript, although data of wind speed and direction is mentioned in the section 2.2(Line 31), the analysis regarding this data was not provided in the manuscript. In addition to the wind, precipitation could also be a controlling factor. Further analysis of the other factors should also be provided in the manuscript.

Reply: We have considered precipitation and wind speed as climatic factors affecting PM10 concentration and dust deposition. The linkage between precipitation and wind speed and PM10 concentration and dust deposition are discussed in first and second paragraph of Section 3.4 in revised manuscript.

Minor comments

Question 1: Page 1, Line 17 : : :(particulate matter 10 m in aerodynamic diameter): : : Please rephrase the sentence in the parentheses.

Reply: Revised as "particulate matter in aerodynamic diameter $\leq 10~\mu$m".

Question 2: Page 1, Line 26-27 : : :The arid climate likely influenced the high dust

deposition and PM10 concentration in the region: : : This sentence is uncorroborated by the manuscript.

Reply: This sentence was deleted.

Question 3: Page 1, Line 29 This study suggests that sand storms are a major factor affecting: : : Please change "are" to "is".

Reply: This sentence was revised as "This study suggests that sand storm is a major factor affecting the temporal variability and spatial distribution of dust deposition in northwest China."

Question 4: Page 2, Line 7-8: An understanding of atmospheric dust sources, emissions, and deposition is therefore essential to improve regional air quality. This sentence is not logically related to the information given before it. The discussion prior to it can't lead to the conclusion that this kind understanding can be helpful to improve regional air quality.

Reply: This sentence was revised as "An understanding of atmospheric dust sources, emissions, and deposition is therefore essential to improve our knowledge of dust impact on regional air quality".

Question 5: Page 2, Line 30 : : : that spans the 21st century. The sentence is overstated, since only 2000-2013 was analyzed in the study, which certainly did not span 21st century.

Reply: The description of "that spans the 21st century" has been deleted.

Question 6: Page 3, Line 31 Daily meteorological data, including surface wind speed and direction : : : Even though the surface wind speed and direction are mentioned in the data description, the analyses relating to them are not given in the manuscript.

Reply: This sentence was revised as "Daily meteorological data including dust days, surface wind speed and precipitation, were collected from the China Meteorological

Administration". We do include information regarding wind speeds in Section 3.4 in revised manuscript.

Question 7: Page 5, Line 2 This industrial belt includes Changji and Urumqi. High dust deposition in the industrial belt was due to industry, coal burning and vehicle exhaust. This explanation is possible, with the anthropogenic source of dust is considered. Please further strengthen this conjecture with relevant papers. In addition to the Changji and Urumqi, Hami, which is also a city located in northern Xinjiang, also had a high dust deposition value as depicted in Figure.2. Why?

Reply: We revised it as "This industrial belt includes Changji and Urumqi. High dust deposition in the industrial belt was due to local industry, coal burning and vehicle exhaust (Matinmin and Meixner, 2011; Zhang et al., 2014b). Therefore, the mixing of the anthropogenic aerosol with transported desert dust contributed to deposition in Changji and Urumqi (Li, et al., 2008)." New references have been included in this section as:

Li, J., Zhuang, G., Huang, K., Lin, Y., Xu, C., and Yu, S.: Characteristics and sources of air-borne particulate in Urumqi, China, the upstream area of Asia dust, Atmospheric Environment, 42(4), 776-787, doi:10.1016/j.atmosenv.2007.09.062, 2008.

Mamtimin, B., and Meixner, F.: Air pollution and meteorological processes in the growing dryland city of Urumqi (Xinjiang, China), Science of the Total Environment, 409, 1277-1290, doi:10.1016/j.scitotenv.2010.12.010, 2011.

Zhang, X.X., Chen, X., Guo, Y.H., Wang, Z.F., Liu, L.Y., Paul, C., Li, S.Y., and Pi, H.W.: Ambient TSP concentration and dustfall variation in Urumqi, China, Journal of Arid Land, 6(6), 668-677, doi:10.1007/s40333-014-0069-6, 2014b.

Hami is located in eastern Xinjiang Province. The city has a population of over 0.5 million and lacks industry characteristic of Changji and Urumqi. The high dust deposition at Hami is due to dust storms originating in the Turpan Depression, not industry.

Question 8: Page 6, Line 17 .. data suggest that particulate matter is the main air pollutant in Xinjiang. The PM10 constituent accounted for 48.7% and 48.2% of the API in the Kuytun and Urumqi. It is necessarily suggest the particulate matter is the main air pollutant?

Reply: Yes! We re-checked the API daily data of the six selected stations (see section 3.3). "The PM10 constituent accounted for 48.7%, 78.4%, 96.2%, 91.5%, 99.5%, and 99.6% of the API in the respective above cities." As for Kuytun, excellent air quality (API<50) accounted 51.3% (Fig.8), therefore, particulate matter is the main air pollutant.

Question 9: Page 6, Line 31-Page 7, Line 1-11 This decline in dust deposition or PM10 concentration could be due to a decrease in frequency of severe dust days versus frequency of dust days from 2000 to 2013 in the region: : :. Nevertheless, in examining the relationship between average annual dust days and dust deposition or PM10 concentration across stations, the frequency of dust days was closely related to dust deposition (R2=0.93) (Fig.10) and ambient PM10 concentration (R2=0.89) (Fig.11). There was a significant 10 increase in dust deposition (7.91 t km-2 day-1) and PM10 concentration (2.06 g m-3 day-1) associated with an increase in dust days. In this paragraph, the relationship between dust deposition/PM10 concentration and dust day frequency at each station is investigated. The result, admittedly, is evident show there is a connection. According to the definition of different dust days, which can be found in section 2.2(page4, line1-5), blowing dust and dust storm constitutes days in which dust is emitted at the station, while dust-in-suspension constitutes days in which dust is not emitted at the station. However, the scatter plot fails to distinguish the inherent difference between these three dust events. Moreover, since the dust is not emitted at this station during dust-in-suspension days, the conclusion given by author, that there appeared to be a close association between frequency of dust-in-suspension events and dust deposition, become unconvincing.

Reply: We examined trends in dust-in-suspension, blowing dust, and dust storms from

2000-2013. Note that there was no trend in blowing dust or dust storms from 2000 to 2013. Over this same time period, there was a significant decrease in frequency of dust-in-suspension. Although we did not show these trends in the paper, the trend for fewer dust-in-suspension coincides with the decline in dust deposition and PM10 concentration from 2000 to 2013. Based upon similarity in trends, there appeared to be some connection between dust-in-suspension and dust deposition.

Question 10: In page 10, Figure 2, please add units for dust deposition in the legend within the figure. In page 17, Figure 5, please add units for PM10 concentration in the legend within the figure.

Reply: We added units for dust deposition and PM10 concentration in Figure 2 and Figure 5 in revised manuscript.

Please also note the supplement to this comment:
http://www.atmos-chem-phys-discuss.net/acp-2016-512/acp-2016-512-AC1-
supplement.pdf

[revised manuscript text omitted]

---

## Referee Comment (RC2) · Anonymous Referee #2 · 22 Oct 2016

The authors analyze dust deposition and concentration data collected in the Xinjiang Province in northwest China over a time period of 13 years. Spatio-temporal differences in annual and 13-year averages for 14 stations are investigated.

The manuscript is overall well written and organized and the presented data and analysis are valuable and interesting. However, my impression is that interpretation of the data is relatively shallow and I think that further in-depth analyses, background information, and more detailed interpretation would be needed before the manuscript can be considered for publication in ACP. Please see my specific comments below.

1) P. 2, l. 10-11: The authors motivate their study by stating that Pye (1987) suggested a lack in reliable dust deposition data. This reference is almost 30 years old. I

[Figure]

would suspect that more data has been collected since. In fact, the authors list several newer references for data on dust deposition in their Table 1. In my opinion a more comprehensive overview and discussion on currently available dust deposition data is needed.

2) P. 2, l. 25-26: What are the processes governing dust emissions, transport, and deposition in Asia? And what data was used by Shao et al. (2011) and Groll et al. (2013) as the basis of their findings (note that the reference should be Shao et al. (2011) rather than Shao (2011))? I think these questions need more attention in the paper, especially to support interpretation of the spatial and temporal variability observed in Xinjiang Province. How is the data presented in the manuscript on hand different/better than the data used in earlier studies? What drives the trends and spatial variability? These questions appear mostly unanswered in the present paper.

3) Are the 14 environmental monitoring stations the same as the air-quality monitoring stations where the API is obtained from?

4) P. 4, l. 3: "Dust-in-suspension constitutes days...". Present weather reports refer to the time of observation, not to the whole day. What category is used for a day in p. 7, l.4 if the 3-hourly data shows two different reports on the same day, e.g. blowing dust and dust in suspension? The authors have chosen to not show the results of their present weather report analysis (p. 7, l. 6). However, it would be interesting to see the outcome and compare to earlier studies using a similar method (see first reference in my comment 8).

5) P. 5, l. 2: It is stated that high dust depositions in the industrial belt were caused by "industry, coal burning and vehicle exhaust". What are the underlying references or data used to support this statement? Or is this a hypothesis? Does the API data used later in the paper provide any evidence in that regard?

6) P. 5, l. 31: Based on their data, the authors "suggest a positive relationship between dust deposition and PM10 concentration". This is to be expected as – apart

from wet scavenging through precipitation – dust deposition is caused by gravitational settling and turbulent diffusion. Both processes are dependent on dust concentration, i.e. the higher the dust concentration, the higher the dust deposition. I think a more detailed discussion of the observation results on the basis of the physics underlying dust deposition would be needed here. Correspondingly, I would suggest to present dust deposition as a function of PM10 concentration in Fig. 7 (and discuss results accordingly) rather than vice versa.

7) P. 6, l. 20-21: The authors state that "weather appears to be a dominant factor" driving dust concentration and deposition in arid regions. This is very vague and only very few details are discussed in the following. It seems clear that atmospheric and land-surface conditions are decisive for local dust entrainment and that atmospheric flow determines dust transport. A more detailed discussion of the predominant regional circulations in Xinjiang province would also help interpretation of the spatio-temporal variability of dust deposition in the area.

8) Section 3.4: How do the results obtained in this paper (e.g. trends) compare to earlier studies on dust variability in central Asia (see for example Shao et al. (2013, doi:10.1002/jgrd.50836) and Xi and Sokolik (2015, doi:10.1002/2015JD024092)). I think more consideration need to be given to previous works, even though they might not deal with the exact same area.

9) P. 7, l. 19-20: "These results suggest that dust source[s] in central Asia affect regional air quality and [are] a potential contributor of global dust." Other studies (e.g. Shao et al., 2011, doi: 10.1016/j.aeolia.2011.02.001, and references therein; Huneeus et al., 2011, doi: 10.5194/acp-11-7781-2011; Ginoux et al., 2012, doi: 10.1029/2012RG000388) have shown that some of the world's major dust sources are located in central Asia. Please rephrase the statement so that it becomes clear in what way the present results support earlier findings, and it what way they may differ.

10) P. 7, l. 31: ". . .this work can aid in adjusting model parameters. . .". While measured

dust depositions can certainly be used to evaluate dust model outputs, it does not seem that this is a direct follow-up on the present manuscript or an objective of this study. In my opinion, this work can rather – if further detailed discussions are added – support understanding of dust deposition along with its spatio-temporal variability in the study area (of course this could then also support model development and evaluation in the future) and I would suggest to motivate the paper as such.

---

## Author Comment (AC2) · 11 Nov 2016

We thank anonymous referee #2 for his/her supportive and thoughtful remarks.

Anonymous Referee #2

Comments

Question 1: P. 2, l. 10-11: The authors motivate their study by stating that Pye (1987) suggested a lack in reliable dust deposition data. This reference is almost 30 years old. I would suspect that more data has been collected since. In fact, the authors list several newer references for data on dust deposition in their Table 1. In my opinion a more comprehensive overview and discussion on currently available dust deposition

data is needed.

Reply: We have revised the paper with a more comprehensive overview and discussion on currently available dust deposition data in the Introduction (Page 2, Line 11-25). New citations have been added to the Introduction to support the above discussion. Citations added to the Introduction include:

Duce, R.A., Liss, P.S., Merrill, J.T., Atlas, E.L, Buat-Menard, P., Hicks, B.B., Miller, J.M., Prospero, J.M., Arimoto, R, Church, T.M., Ellis, W., Galloway, J.N., and Hansen, L.: The atmospheric input of trace species to the world ocean, Global Biogeochem. Cycles, 5, 193-259, doi: 10.1029/91GB01778 , 1991.

Ginoux, P., Chin, M.. Tegen, I., Prospero, J.M., Holben, B., Dubovik, O., and Lin, S.: Sources and distributions of dust aerosols simulated with the GOCART model, Journal of Geophysical Research, 106(17), 20255-20273, doi: 10.1029/2000JD000053, 2001. Ginoux, P., Prospero, J.M., Gill, T.E., Hsu, N.C., and Zhao, M.: Global-scale attribution of anthropogenic and natural dust sources and their emission rates based on MODIS Deep Blue aerosol products, Reviews of Geophysics, doi: 10.1029/2012RG000388, 2012.

Huneeus, N., Schulz, M., Balkanski, Y., Griesfeller, J., Prospero, J., Kinne, S., Bauer, S., Boucher, O., Chin, M., Dentener, F., Diehl, T., Easter, R., Fillmore, D., Ghan, S., Ginoux, P., Grini, A., Horowitz, L., Koch, D., Krol, M. C., Landing, W., Liu, X., Mahowald, N., Miller, R., Morcrette, J.J., Myhre, G., Penner, J., Perlwitz, J., Stier, P., Takemura, T., and Zender, C. S.: Global dust model intercomparison in AeroCom phase I, Atmos. Chem. Phys., 11, 7781-7816, doi:10.5194/acp-11-7781-2011, 2011.

Mahowald, N.M, Kohfeld, K.E., Hansson, M., Balkanski, Y., Harrison, S.P., Prentice, I.C., Michael, S., and Rodhe, H.: Dust sources and deposition during the last glacial maximum and current climate: a comparison of model results with paleodata from ice cores and marine sediments, J. Geophys. Res. 104, 15895-916, 1999.

Mahowald, N.M., Engelstaedter, S., Luo, C., Sealy, A., Artaxo, P., Benitez-Nelson, C., Bonnet, S., Chen, Y., Chuang, PY., Cohen, DD., Dulac, F., Herut, B., Johansen, A.M., Kubilay, N., Losno, R., Maenhaut, W., Paytan, A., Prospero, JM., Shank, L.M., and Siefert, R.L.: Atmospheric iron deposition: global distribution, variability, and human perturbations, Annu. Rev. Marine. Sci, 1, 245-278, doi: 10.1146/an-nurev.marine.010908.163727, 2009.

Prospero J.M.: Long-range transport of mineral dust in the global atmosphere: impact of African dust on the environment of the southeastern United States, Proc. Natl. Acad. Sci. 96, 3396-3403, doi: 10.1073/pnas.96.7.3396, 1999.

Zhang, X.Y., Arimoto, R., and An, Z.S.: Dust emission from Chinese desert sources linked to variations in atmospheric circulation, 102(D23), 28041-28047, doi:10.1029/97JD02300, 1997.

Question 2: P. 2, l. 25-26: What are the processes governing dust emissions, transport, and deposition in Asia? And what data was used by Shao et al. (2011) and Groll et al. (2013) as the basis of their findings (note that the reference should be Shao et al. (2011) rather than Shao (2011))? I think these questions need more attention in the paper, especially to support interpretation of the spatial and temporal variability observed in Xinjiang Province. How is the data presented in the manuscript on hand different/better than the data used in earlier studies? What drives the trends and spatial variability? These questions appear mostly unanswered in the present paper.

Reply: The Eurasian atmospheric circulation is governing dust emissions, transport, and deposition in central Asia. Compared with earlier studies, the data presented in this manuscript were observed at 14 environmental stations in Xinjiang, northwest China during 2000-2013 with a monthly temporal resolution, which would be helpful to improve our knowledge of dust impact on regional air quality. The 14-year continuous deposition data was collected according to Chinese national standards, which fills the gap in the central Asian arid region where observations are scarce. The atmospheric

circulation such as cyclones (Shao et al., 2013) primarily drives and influences the trends and spatial variability of dust deposition and ambient PM10 concentration. We added this discussion in Section 3.4.

Question 3: Are the 14 environmental monitoring stations the same as the air-quality monitoring stations where the API is obtained from?

Reply: Yes, the 14 environmental monitoring stations are the same as the air-quality monitoring stations where the API is obtained from.

Question 4: P. 4, l. 3: "Dust-in-suspension constitutes days: : :". Present weather reports refer to the time of observation, not to the whole day. What category is used for a day in p. 7, l.4 if the 3-hourly data shows two different reports on the same day, e.g. blowing dust and dust in suspension? The authors have chosen to not show the results of their present weather report analysis (p. 7, l. 6). However, it would be interesting to see the outcome and compare to earlier studies using a similar method (see first reference in my comment 8).

Reply: We used the most severe dust category if two or more observations were made on a single day. For example, if both blowing dust and dust in suspension were observed at one meteorological station during a single day, we categorized this event as a blowing dust day. A comparison of our method to methods used in earlier studies is provided in Fig. S1 (in the Supplement).

Question 5: P. 5, l. 2: It is stated that high dust depositions in the industrial belt were caused by "industry, coal burning and vehicle exhaust". What are the underlying references or data used to support this statement? Or is this a hypothesis? Does the API data used later in the paper provide any evidence in that regard?

Reply: We revised the text as "This industrial belt includes Changji and Urumqi. High dust deposition in the industrial belt was due to local industry, coal burning and vehicle exhaust (Matinmin and Meixner, 2011; Zhang et al., 2014b). Therefore, the mixing

of the anthropogenic aerosol with transported desert dust contributed to deposition in Changji and Urumqi (Li, et al., 2008)." Both Fig. 2 and 8 (API data used later in the paper) provide evidence that dust deposition in Changji and Urumqi is due to industry or vehicles. New references have been added to this section as:

Li, J., Zhuang, G., Huang, K., Lin, Y., Xu, C., and Yu, S.: Characteristics and sources of air-borne particulate in Urumqi, China, the upstream area of Asia dust, Atmospheric Environment, 42(4), 776-787, doi:10.1016/j.atmosenv.2007.09.062, 2008.

Mamtimin, B., and Meixner, F.: Air pollution and meteorological processes in the growing dryland city of Urumqi (Xinjiang, China), Science of the Total Environment, 409, 1277-1290, doi:10.1016/j.scitotenv.2010.12.010, 2011.

Zhang, X.X., Chen, X., Guo, Y.H., Wang, Z.F., Liu, L.Y., Paul, C., Li, S.Y., and Pi, H.W.: Ambient TSP concentration and dustfall variation in Urumqi, China, Journal of Arid Land, 6(6), 668-677, doi:10.1007/s40333-014-0069-6, 2014b.

Question 6: P. 5, l. 31: Based on their data, the authors "suggest a positive relationship between dust deposition and PM10 concentration". This is to be expected as - apart from wet scavenging through precipitation – dust deposition is caused by gravitational settling and turbulent diffusion. Both processes are dependent on dust concentration, i.e. the higher the dust concentration, the higher the dust deposition. I think a more detailed discussion of the observation results on the basis of the physics underlying dust deposition would be needed here. Correspondingly, I would suggest to present dust deposition as a function of PM10 concentration in Fig.7 (and discuss results accordingly) rather than vice versa.

Reply: We have briefly discussed the physics of dust deposition in Section 3.2 in the revised manuscript (Page 6, Line 9-16). Moreover, we changed the x-axis and y-axis of Fig. 7 to present dust deposition as a function of PM10 concentration, and added discussion in this section.

Question 7: P. 6, l. 20-21: The authors state that "weather appears to be a dominant factor" driving dust concentration and deposition in arid regions. This is very vague and only very few details are discussed in the following. It seems clear that atmospheric and land-surface conditions are decisive for local dust entrainment and that atmospheric flow determines dust transport. A more detailed discussion of the predominant regional circulations in Xinjiang province would also help interpretation of the spatio-temporal variability of dust deposition in the area.

Reply: A more detailed discussion of the predominant regional circulation in Xinjiang province has been added in Section 3.4 (Page 7, Line 5-9) to interpret the spatio-temporal variability of dust deposition in the study area.

Question 8: Section 3.4: How do the results obtained in this paper (e.g. trends) compare to earlier studies on dust variability in central Asia (see for example Shao et al. (2013, doi:10.1002/jgrd.50836) and Xi and Sokolik (2015, doi:10.1002/2015JD024092)). I think more consideration need to be given to previous works, even though they might not deal with the exact same area.

Reply: The results obtained in this paper are based on the measurement from observation, which reflect the regional dust characteristics. The high PM10 concentration in 2001 and 2002 over study region (Fig. 6) is in accordance with Shao et al. (2013) as described "years of high dust activities in east Asia". Moreover, the decreasing trend of PM10 concentration and dust deposition in the study is consistent with Shao et al. (2013) as described "declining dust activities in east Asia since the late 1970s". Further consideration of earlier studies has been given, and we added the following citations to the discussion section:

Shao, Y., Kolse, M., and Wyrwoll, K.: Recent global dust trend and connections to climate forcing, Journal of Geophysical Research, doi: 10.1002/jgrd.50836, 2013.

Xi, X., and Sokolik, I.N.: Dust interannual variability and trend in Central Asia from 2000 to 2014 and their climatic linkages, Journal of Geophysical Research, doi:

10.1002/2015JD024092, 2015.

Question 9: P. 7, l. 19-20: "These results suggest that dust source[s] in central Asia affect regional air quality and [are] a potential contributor of global dust." Other studies (e.g. Shao et al., 2011, doi: 10.1016/j.aeolia.2011.02.001, and references therein; Huneeus et al., 2011, doi: 10.5194/acp-11-7781-2011; Ginoux et al., 2012, doi: 10.1029/2012RG000388) have shown that some of the world's major dust sources are located in central Asia. Please rephrase the statement so that it becomes clear in what way the present results support earlier findings, and it what way they may differ.

Reply: In above papers by Shao et al., 2011; Huneeus et al., 2011; and Gnioux et al., 2012; they suggested that uncertainty in dust deposition is an important problem in current research because of a limited number of observations. This uncertainty severely influences the accurate estimation from models. Our observation confirmed that the study area is a potential dust source region as described by Shao, et al. (2011) and Ginoux et al. (2012). We rephrase the statement in the Conclusion (Page 8, Line 19-21).

Question 10: P. 7, l. 31: ": : :this work can aid in adjusting model parameters: : :". While measured dust depositions can certainly be used to evaluate dust model outputs, it does not seem that this is a direct follow-up on the present manuscript or an objective of this study. In my opinion, this work can rather – if further detailed discussions are added – support understanding of dust deposition along with its spatio-temporal variability in the study area (of course this could then also support model development and evaluation in the future) and I would suggest to motivate the paper as such.

Reply: We're now preparing the next manuscript of modeling on dust deposition in the study area.

Please also note the supplement to this comment:

http://www.atmos-chem-phys-discuss.net/acp-2016-512/acp-2016-512-AC2-supplement.pdf

[Figure]

- Y-axis: Dust deposition(t·km$^{-2}$), ranging from 0 to 1400
- X-axis: PM$_{10}$ concentration(μg·m$^{-3}$), ranging from 0 to 300

Equation on plot:
$$y=1.46\ln x-1.48$$
$$R^2=0.81$$

**Fig. 1.** Figure 7. Relationship between annual dust deposition and PM10 concentration in Xinjiang Province. Each point represents data averaged across 2000 to 2013 at one station.

---

## Author Response (AR2)

**Dust deposition and ambient PM$_{10}$ concentration in northwest China: Spatial and temporal variability**

Xiao-Xiao Zhang, Brenton Sharratt, Xi Chen, Zi-Fa Wang, Lian-You Liu, Yu-Hong Guo, Jie Li, Huan-Sheng Chen, and Wen-Yi Yang

We thank anonymous referee for his/her supportive and precious overview remarks.

**Comments**

*Question 1:* A problem is however that the flow pattern in the Tarim Basin is not analysed. This is problematic, because dust distribution in the basin is strongly affected by the flow pattern in the basin. We know that the high dust concentration in the southern part of the basin is due to dune transport and convective activities. As one of the authors, e.g., Prof. Zifa Wang, has done a lot of air quality simulations, I would guess that he knows well about the flow pattern in the basin. Related to this is that I find the regression graphs are not so meaningful, because these relationships are not anything universal. In general, I wish the paper can be written more concisely.

*Reply:* We've added further discussion on wind flow patterns in the Tarim Basin on page 8 (line 2-10) as follows:

Dust distribution in south Xinjiang (including the Tarim Basin and Taklamakan Desert), however, is strongly affected by wind flow patterns. Aeolian transport in the Taklamakan Desert is predominantly from the northeast toward the south (Wang et al., 2014; Rittner et al., 2016). Huang et al. (2014) reported that the Taklamakan Desert is a source of fine dust particles ($\leqslant 3\mu m$ in aerodynamic diameter) which significantly influences East Asia. Strong northeast winds dominate the prevailing wind regime in the eastern Taklamakan Desert; these winds influence air quality in both the eastern and the southeastern parts of the desert. The western and northern parts of the Taklamakan Desert and Tarim Basin are highly affected by west, northwest and north winds (Sun and Liu, 2006; Zan et al., 2014; Li et al., 2015). Under prevailing winds, dust aerosols are transported from northern to the southern Taklamakan Desert (e.g. Hotan city) and thereby cause high ambient PM$_{10}$ concentration and dust deposition.

New citations have been added to the Reference as follows:

Li, X., Feng, G., Sharratt, B., and Zheng, Z.: Aerodynamic properties of agricultural and natural surfaces in northwestern Tarim Basin, Agricultural and Forest Meteorology, 204, 37-45, doi:10.1016/j.agrformet.2015.01.005, 2015.

Rittner, Martin., Vermeesch, Pieter., Carter, Andrew., Bird, Anna., Stevens, Thomas., Garzanti, Eduardo., Andò, Sergio., Vezzoli, Giovanni., Dutt, Ripul., Xu, Zhiwei., and Lu, Huayu.: The provenance of Taklamakan desert sand, Earth and Planetary Science Letters, 437, 127-137, doi: 10.1016/j.epsl.2015.12.036, 2016.

Sun, J., and Liu, T.: The age of the Taklimakan Desert, Science, 312(5780), 1621, doi: 10.1126/science.1124616, 2006.

Wang, Haibing., Jia, Xiaopeng., Lib, Kuan., and Wang, Hongfang.: External supply of dust in the Taklamakan sand sea, Northwest China, reveals the dust-forming processes of the modern sand sea surface, CATENA, 119, 104-115,

doi:10.1016/j.catena.2014.03.015, 2014.

Zan, Jinbo., Fang, Xiaomin., Appel, Erwin., Yan, Maodu., and Yang, Shengli.: New insights into the magnetic variations of aeolian sands in the Tarim Basin and its paleoclimatic implications, Physics of the Earth and Planetary Interiors, 229, 82-87, doi:10.1016/j.pepi.2014.01.010, 2014.

We have modified the regression formula in Figure 7, which may be of future value in assessing the global dust deposition.

[Figure]

**Figure 7.** Relationship between annual dust deposition and PM$_{10}$ concentration in Xinjiang Province. Each point represents data averaged across 2000 to 2013 at one station.

*Question 2:* Xinjiang Province is not the correct Chinese expression

*Reply:* Xinjiang Uygur Autonomous Region is the official expression. We noticed that both of these expressions (Xinjiang Province and Xinjiang Uygur Autonomous Region) are adopted in several scientific research papers; the former expression can be found in:

Li, X., Feng, G., Sharratt, B., and Zheng, Z.: Aerodynamic properties of agricultural and natural surfaces in northwestern Tarim Basin, Agricultural and Forest Meteorology, 204, 37-45, doi: 10.1016/j.agrformet.2015.01.005, 2015. (expressed in "Xinjiang Province")

Sullivan R.C. et al.: Direct observations of the atmospheric processing of Asian mineral dust, Atmos. Chem. Phys., 7, 1213-1236, 2007. (expressed in "Xinjiang Province")

Hu X,Q., et al.: Operational retrieval of Asian sand and dust storm from FY-2C geostationary meteorological satellite and its application to real time forecast in Asia, Atmos. Chem. Phys., 8, 1649-1659, 2008. (expressed in "Xinjiang Province")

Therefore, we used the expression in the manuscript.

***Question 3:*** Introduction is somewhat weak. More focus should be on the scientific problem

***Reply:*** We've revised and reinforced the Introduction (page 2, line 17-30).

***Question 4:*** P4, L13, cite the WMO document

***Reply:*** A WMO document link (http://www.wmo.int/pages/prog/www/WMOCodes.html) has been cited in page 4, line 17.

***Question 5:*** Which areas are north, east and south Xinjiang?

[revised manuscript text omitted]